# The Use of Mefenoxam to Treat Cutaneous and Gastrointestinal Pythiosis in Dogs: A Retrospective Study

**DOI:** 10.3390/microorganisms11071726

**Published:** 2023-06-30

**Authors:** Phillip Billings, Stuart Walton, Justin Shmalberg, Domenico Santoro

**Affiliations:** 1Department of Small Animal Clinical Sciences, College of Veterinary Medicine, University of Florida, 2015 SW 16th Avenue, Gainesville, FL 32610, USA; phbillin@ufl.edu (P.B.); stuartawalton@ufl.edu (S.W.); 2Department of Comparative, Diagnostic and Population Medicine, College of Veterinary Medicine, University of Florida, Gainesville, FL 32610, USA; shmalberg@ufl.edu

**Keywords:** *Pythium insidiosum*, oomycetes, oomycosis, mefenoxam, pythiosis, cutaneous and gastrointestinal

## Abstract

*Pythium insidiosum*, an aquatic oomycete with pathogenic potential in mammals, causes gastrointestinal and cutaneous disease in dogs. Mefenoxam, an agricultural anti-oomycotic compound, has a demonstrated the ability to inhibit *P. insidiosum* growth in vitro and has been associated with efficacy in treating gastrointestinal pythiosis in several case reports. Electronic medical records of dogs seen at University of Florida Small Animal Hospital and treated with mefenoxam between 2013 and 2020 were searched. Dogs were included in this study upon previous definitive diagnosis with either organism identification using culture, PCR, or antibody ELISA, or a combination of these tests with or without supportive histopathological analysis. Since 2013, mefenoxam had been administered to 25 dogs with cutaneous pythiosis and 16 dogs with gastrointestinal pythiosis. In both gastrointestinal and cutaneous pythiosis groups, the administration of mefenoxam was associated with a survivability rate of approximately 51%. There was a statistically significant difference in the time to death between cutaneous (245 days (52–530)) and gastrointestinal (90 days (21–203)) groups; dogs infected with cutaneous pythiosis survived significantly longer after being diagnosed with the disease (*p* = 0.035). The dogs in this study experienced increased survival rates and time to death, in the absence of side effects due to mefenoxam, compared with previously published literature. The results of this retrospective study, with some limitations, are promising and should prompt further investigation into the use of mefenoxam in the treatment of both gastrointestinal and cutaneous pythiosis.

## 1. Introduction

*Pythium insidiosum*, an infective aquatic oomycete, is gaining more recognition in veterinary medicine due to its expanding environmental niche [1]. The preferred habitat of this microorganism is warm, stagnant water in tropical and subtropical climates, such as Southeast Asia, coastal Australia, and the southeastern United States, but can also occur in more temperate regions [1]. This pseudo-fungus causes pythiosis, which commonly manifests as cutaneous disease in dogs, cats, horses, and cattle, as well as humans, and also as gastrointestinal disease in dogs [2]. Disease is presumptively acquired through direct contact with water containing zoospores, which adhere to and encyst in damaged skin or intestinal mucosa [1]. Clinical signs associated with gastrointestinal pythiosis include weight loss, diarrhea, and anorexia, with possible extension to local mesenteric lymph nodes [3]. Dermatological disease manifests as cutaneous and subcutaneous ulcerative nodules and draining tracts [4]. 

Enzyme-linked immunosorbent assay (ELISA) antibody detection, immunohistochemistry, polymerase chain reaction (PCR), and *P. insidiosum* isolation in culture are currently used for the definitive diagnosis of pythiosis [2]. Conclusive diagnostic tests are essential to differentiate pythiosis from other oomycete (e.g., lagenidiosis) and mucormycete (e.g., zygomycosis) infections [2]. Prognosis in dogs is guarded due to early misdiagnosis of disease and variable efficacy of currently available treatments [1,2,3,5,6,7].

Currently, there is no consensus on the “gold standard” treatment for pythiosis, with aggressive surgical resection being recommended whenever possible, followed by adjunctive long-term medical therapy to control microscopic disease and prevent recurrence [1,2,4,5]. Adjunctive medical therapy historically consists of systemic antifungals variably combined with prednisone. However, the reported efficacy of antifungals is poor, with approximately 20% of dogs responding positively to the use of itraconazole, terbinafine, and amphotericin B [5]. This variable efficacy is due to *P. insidiosum’s* cell wall containing cellulose and β-glucan but its plasma membrane lacking ergosterol, a common target for antifungals [1,2].

Because of the lack of definitive, predictable, and effective treatment options, alternative therapies have been sought in the treatment of pythiosis. These include minocycline alone and in conjunction with immunotherapy [8]; other antifungal azoles (e.g., voriconazole and posaconazole) and caspofungin, which have displayed some in vitro inhibitory activity against *P. insidiosum* [9]; and photodynamic therapy, which significantly inhibits *P. insidiosum* growth in culture [10]. Unfortunately, some therapeutic treatments effective in other animal species do not have the same efficacy in dogs (e.g., immunotherapeutic injections used in horses have not demonstrated the same efficacy in dogs) [6,11]. Other adjunctive therapies, such as hyperbaric oxygen therapy, have been successfully used in combination with both surgical and medical therapy to improve tissue oxygenation and decrease microbial spread in patients with cutaneous oomycosis [12].

Mefenoxam, an agricultural antifungal and anti-oomycotic compound, has gained popularity as a potential breakthrough therapy in the treatment of pythiosis. This compound blocks RNA polymerase impairing rRNA synthesis in fungal zoospores [13]. At concentrations of 1 μg/mL, mefenoxam inhibits over 90% of *P. insidiosum* proliferation in vitro [5,9]. Within the veterinary literature, several case reports and case studies have reported the successful use of mefenoxam either alone or in combination with aggressive surgical resection and other adjunctive medical therapies, such as itraconazole and terbinafine [4,5,14]. At the currently suggested dose of 8 mg/kg/day, no adverse side effects with prolonged long-term use have been reported in dogs [15]. Additionally, an oral administration study in laboratory rats identified an LD_50_ of 490 mg/kg, suggesting the compound’s favorable safety profile in other mammalian species [9]. Currently, there is a paucity of information with regards to both the efficacy and utility of mefenoxam in the treatment of canine pythiosis. Furthermore, there is a genuine need for additional information in support of the use of this compound in clinical practice. Thus, the purpose of this study was to describe the use of mefenoxam, either alone or in combination with other therapies, to treat canine pythiosis over a seven-year period at a specialty referral veterinary hospital in Florida. 

## 2. Materials and Methods

Electronic medical records (Cornerstone^®^ Veterinary Software, IDEXX^®^, version 9.4) of all dogs diagnosed with pythiosis and treated with mefenoxam were searched between the years 2013 and 2020 at the tertiary referral center. Visit records were manually reviewed, and data were retrieved. Dogs with a diagnosis of gastrointestinal and/or cutaneous pythiosis treated with mefenoxam oral solution (Subdue Maxx^®^, Syngenta Crop Protection, LLC, Greensboro, NC, USA) for more than three days were included in this study. Historically, each owner had signed a consent form stating that they were consenting to the use of mefenoxam, that the agent was not an approved drug, and that the side effects were unknown. Additionally, all owners consented to euthanasia if the disease proved progressive or if the side effects from mefenoxam affected the quality of life. A definitive diagnosis of pythiosis was achieved with PCR [16], antibody ELISA (>40% seropositivity) [17], isolation of *P. insidiosum* in culture, or a combination of these tests with or without histopathological analysis. 

Retrieved data from medical records included breed, age, sex, weight (for dogs in the gastrointestinal group), exposure to stagnant water, and lifestyle. Information specific to pythiosis included clinical signs (e.g., poor body condition score, diarrhea, cutaneous nodules and ulcers, etc.), length of symptoms before treatment, diagnostic tests performed, whether surgical resection was performed, duration of mefenoxam treatment, dose of mefenoxam, and length of time to cure and/or death. Owners and/or referring veterinarians were contacted if records did not indicate a cure or death. Cured was defined as resolution of clinical signs and/or negative serum ELISA test. A time to cure was only recorded in dogs with an ELISA no longer supportive of active pythiosis or in dogs that had an examination at the study institution. Dogs off medications for at least 90 days, free of clinical signs and active lesions, and reportedly normal according to the owner or referring veterinarian were considered cured, but without a definitive time to cure. The use of other therapies was also recorded and is displayed in Table 1. Dogs were labeled as cured, dead, lost to follow-up, or dead from inconclusive causes based on case outcomes. 

### Statistical Analysis

Descriptive statistic parameters for cutaneous and gastrointestinal pythiosis were reported as averages, standard deviations, medians, and ranges depending on the nature of the data. When appropriate (e.g., age, doses, and duration of clinical signs), the normality distribution was tested using the Shapiro–Wilk test with α = 0.05. Variables between the two groups (cutaneous vs. gastrointestinal) were compared using either the unpaired *t*-test or the Mann–Whitney test, based on the normality test. The Kaplan–Meier survival curve test was used to compare the time to cure and time to death for both cutaneous and gastrointestinal pythiosis. GraphPad 9.1 statistical software was used for analyses (GraphPad Software Inc.; La Jolla, CA, USA). A *p*-value < 0.05 was considered significant. 

## 3. Results

### 3.1. Canine Demographics and Clinical Signs

Forty-five dogs that received mefenoxam during the retrospective period were initially included in the study. Forty-one dogs met the inclusion criteria, with three dogs being excluded because *Lagenidium giganteum* was confirmed post-mortem and one dog was subsequently diagnosed with *Paralagenidium* sp. after tissue PCR analysis. Nine dogs had inconclusive case outcomes, were lost to follow-up, or were cured but without a definitive time to cure. Thus, 16 dogs (n = 11 for cutaneous and n = 5 for gastrointestinal group) were included in the time-to-cure analysis, and 16 dogs (n = 10 for cutaneous and n = 6 for gastrointestinal group) were included in the time-to-death analysis. 

Twenty-five dogs (61%) had cutaneous disease, with the remaining 16 dogs (39%) having gastrointestinal disease. Demographic information is displayed in Table 2. Eleven male (44%) and fourteen female (56%) dogs presented with cutaneous disease. Nine (56%) male and seven (44%) female dogs were diagnosed as having gastrointestinal disease. The median ages of dogs diagnosed with cutaneous and gastrointestinal disease were three (range 0.75–7 years) and two years (range 1–6.5 years), respectively. The two most common breeds represented in both groups were German Shepherd dogs (n = 16; 39%) and Labrador Retriever (n = 6; 15%). Of the German Shepherd dogs, 13 (81%) had cutaneous pythiosis, and 3 (19%) had gastrointestinal pythiosis. Of the six Labrador Retrievers, five (83%) had cutaneous disease, and one (17%) dog was diagnosed with gastrointestinal disease. Other breeds included Siberian Husky, terrier breeds (American pit-bull, American Staffordshire terrier, and American bulldog), and various mixed breed and pure breed dogs, as shown in Table 2. 

Definitive diagnosis was made with ELISA serology, PCR, or isolation of *P. insidiosum* from tissue culture, either as a standalone test (i.e., ELISA, culture, or PCR) or in combination with or without histopathology (e.g., histopathology and pythium culture, pythium culture and PCR, etc.), as reported in Table 2. For the cutaneous group, diagnosis was achieved based on histopathological analysis in 24 dogs (96%), serology ELISA in 21 dogs (84%), tissue culture in 18 dogs (72%), and PCR from tissue culture or biopsy in 13 dogs (52%). Diagnosis of gastrointestinal pythiosis was achieved based on histopathological analysis in 12 dogs (75%), serology ELISA in 12 dogs (75%), PCR from tissue culture or biopsy in 3 dogs (19%), and tissue cultures in 2 dogs (13%), as summarized in Table 2. Twenty (49%) records contained information regarding the dog’s lifestyle, including access to stagnant bodies of water, with seventeen (68%) dogs with cutaneous disease and three (19%) dogs with gastrointestinal disease having access to water. All dogs were from the Southeast, with dogs presenting from Florida (93%), Georgia (5%), and Alabama (2%). 

Clinical signs associated with gastrointestinal disease included a body condition score of <5/9 in 11 dogs (69%), weight loss in 11 dogs (69%), small bowel diarrhea in 11 dogs (69%), vomiting in 9 dogs (56%), anorexia in 8 dogs (50%), and large bowel diarrhea in 7 dogs (44%). Clinical signs associated with cutaneous disease included variably sized cutaneous and subcutaneous nodules (1–15 cm in diameter). Eight dogs (32%) with nodules were reported to be ulcerated, and ten dogs (40%) were reported to have draining tracts. The mean number of gross cutaneous lesions was two (range 1–5). Three dogs (19%) with gastrointestinal disease underwent small intestinal resection and anastomosis, while fifteen cutaneous cases (60%) had surgical resection of grossly infected tissue from varying anatomical locations. 

### 3.2. Use of Mefenoxam

The duration of treatment with mefenoxam varied greatly, with the median time of treatment being 60 days (range 3–1500 days). Two dogs with gastrointestinal disease were euthanized less than a week after beginning mefenoxam due to progression of the clinical signs. One dog with gastrointestinal disease was treated with mefenoxam for more than four years. Mefenoxam was well tolerated with no adverse effects being reported directly from its use.

The median time from diagnosis to beginning treatment for both groups was 45 days (range 10–350 days). All dogs received additional adjunctive therapies, as shown in Table 1. Proportional treatments for each group included prednisone (85% cutaneous and 69% gastrointestinal), hyperbaric oxygen therapy (73% cutaneous and 25% gastrointestinal), minocycline (77% cutaneous and 69% gastrointestinal), and terbinafine (54% cutaneous and 69% gastrointestinal). 

Table 3 shows the case outcomes for both groups following mefenoxam treatment. A total of 21 dogs (n = 13 cutaneous and n = 8 gastrointestinal) were categorized as cured. In total, 16 dogs (n = 10 cutaneous and n = 6 gastrointestinal) died or were euthanized as a result of the disease. A total of 3 dogs (n = 1 cutaneous and n = 2 gastrointestinal) were lost to follow-up and could not be classified as cured or dead. Lastly, one dog in the cutaneous group was euthanized for reasons other than the cutaneous lesions. 

### 3.3. Survivability and Time-to-Cure Assessment

A survival calculation (survivability = cured of disease/[cured of disease + dead from disease + lost to follow-up + inconclusive death]) was derived from the dogs following treatment with mefenoxam. This calculation was used to avoid overestimation of survivability by assuming that those lost to follow-up or had inconclusive deaths died from the disease. This calculation yielded, as shown in Table 3, approximately 52% survivability for the cutaneous and 50% for the gastrointestinal groups. 

There was a statistically significant difference in the time to death between the cutaneous and gastrointestinal groups. Dogs affected by cutaneous pythiosis lived significantly longer than dogs affected by the gastrointestinal form of the disease (Figure 1) (*p* = 0.035; cutaneous median = 245 days (52–530) and gastrointestinal median = 90 days (21–203)). Contrastingly, there was not a statistically significant difference in the time to cure between both groups (Figure 2) (*p* = 0.95; cutaneous median = 105 days (60–360) and gastrointestinal median = 220 days (40–485)) for dogs that responded to mefenoxam therapy. 

### 3.4. Histopathological and Necroscopic Findings 

Post-mortem examination was performed in 7 (17%) dogs (n = 4 cutaneous and n = 3 gastrointestinal). All dogs had evidence of oomycotic infection. Cutaneous cases exhibited dermatitis, panniculitis, and pyogranulomatous or granulomatous eosinophilic inflammation. Gastrointestinal cases displayed mucosal thickening and pyogranulomatous or granulomatous eosinophilic inflammation with pseudo-fungal hyphae throughout affected tissues. Systemic spread of disease was not recognized in any necropsied cases.

## 4. Discussion

This is the first retrospective study to describe the use of mefenoxam in a large cohort of dogs with gastrointestinal or cutaneous pythiosis. Based on our study, mefenoxam given orally in conjunction with other therapeutic interventions resulted in survival times of 245 days (52–530) and 90 days (21–203) for cutaneous and gastrointestinal disease, respectively, without appreciable side effects. Previous publications regarding the use of mefenoxam in cases of canine pythiosis are limited, with treatments being highly variable; therefore, no direct comparisons can be made about survival in the current study and previous case series or single-case reports [4,5,14]. The possibility of synergy with other administered treatment interventions also remains unclear.

Canine cutaneous pythiosis appeared more common than gastrointestinal pythiosis in the examined area. Previously published studies and reviews have described conflicting evidence with regards to the incidence of cases of gastrointestinal and cutaneous pythiosis [1,18]. This could potentially suggest that regionality may play a role in the most common manifestation of this disease seen in dogs. However, there may be other explanations for why more cutaneous cases were seen in this time frame at this hospital. One explanation is that there were a higher percentage of dogs with the cutaneous form being referred due to visualization of the ulcerated cutaneous lesions increasing the suspicion for pythiosis. Confirmatory biopsies and other diagnostic tests are easier in cutaneous cases, and it is possible that gastrointestinal cases are undetected or misdiagnosed in some settings.

A goal of this retrospective evaluation was to show the survivability rate following mefenoxam treatment in large groups of dogs with gastrointestinal or cutaneous pythiosis. In this study, 21 dogs, 13 with cutaneous disease and eight with gastrointestinal disease, had clinical resolution of pythiosis and an overall survivability rate of approximately 51% across both groups. Although it is difficult to compare survival data between case series due to the multitude of factors that could influence case outcomes, the data presented in this study demonstrate that dogs receiving mefenoxam in combination therapy survived longer than dogs historically treated with other treatment options. Indeed, the median survival time previously reported for gastrointestinal pythiosis is 26.5 days (n = 10) [2,6,7] compared with the 90 days (n = 6) reported in the present study. It is also important to note that certain factors, such as adjunctive therapies, stage of disease, and time to diagnosis, may also influence the outcome of the disease and these factors are difficult to examine in a retrospective study.

The difference in time to death between the two groups (Figure 1) could have been due to the advanced nature of gastrointestinal disease at the time of diagnosis, making these cases more challenging to manage medically. Possible explanations for this may include initial failure to identify gastrointestinal pythiosis based upon nonspecific clinical signs (i.e., vomiting, diarrhea, and weight loss), or the advanced state of disease when animals were referred (i.e., poor body condition), or due to the refractory nature of disease with respect to empiric therapies such as intestinal deworming, dietary modification, and antibiotic therapy that may delay treatment. Another possibility may be poor intestinal absorption of the drug due to inflammatory infiltrates. To confirm this hypothesis, further pharmacokinetic evaluation of mefenoxam is needed. Other potential reasons may be associated with an anecdotal poorer prognosis of gastrointestinal pythiosis leading to humane euthanasia.

Although beyond the scope of this study, it is imperative to reiterate the importance of multiple confirmatory diagnostics to make an accurate and definitive diagnosis of canine pythiosis [2]. Four dogs, initially diagnosed with pythiosis, were eventually excluded as the initial diagnosis was solely based on histopathological analysis. A final diagnosis of Lagenidiosis and Paralagenidiosis was made based on a combination of histopathology and tissue PCR at necropsy. The authors recommend that paired diagnostic tests be performed to definitively diagnose this disease. Tissue PCR paired with ELISA, or histopathology paired with PCR, provides greater sensitivity and specificity, because histopathology and/or ELISA alone cannot distinguish *P. insidiosum* from other oomycotic and other related organisms, thus making it a less specific diagnostic modality. Historically, cytology has proved not to be a reliable diagnostic modality for pythiosis, with previous studies revealing hyphae in just 45% of infected samples [19]. However, there is potential for cytology to be a more reliable and less invasive diagnostic tool when it is combined with panfungal PCR assays performed on cytology slides when abundant organisms are present [20]. As with every diagnostic modality, there is a chance for false-positive or false-negative results. Given the timeline analyzed in this study (seven years), some of the diagnostic methods may have been improved to increase accuracy of detection. However, it is important to note that most of the dogs in this study received combinations of multiple diagnostic tests to decrease the likelihood of an improper diagnosis.

*Pythium insidiosum* is known to commonly reside in warm, stagnant waters in the North and Central Florida area [21], which could have been a potential exposure site for many of the dogs in this study. While there was not consistent documentation regarding access to lakes or ponds, many of the owners did report that their dog “enjoyed swimming” or had “free range access” to bodies of water. German Shepherd dogs and Labrador Retrievers were the most common breeds represented in this study. These are two popular large breed dogs that are known to enjoy outdoor recreational activities. In addition, the German shepherd dog has been previously documented for its increased susceptibility to fungal infections, which could potentially raise its vulnerability to other fungal-like infections, such as *P. insidiosum* [22]. The German shepherd dog breed has also been shown to have a relative deficiency in serum immunoglobulin A compared with groups of other healthy dog breeds [23], which could increase the likelihood of this breed to develop disease if exposed to *P. insidiosum;* however, a definitive cause of the suggested increased susceptibility of this breed to pythiosis has not been established.

There is no consensus on the optimal treatment of canine pythiosis with previous recommendations calling for aggressive surgical resection, when possible, and follow-up adjunctive treatments, such as antifungals, antibiotics, and immunotherapeutics [1,2,4,5,24]. Since medical therapy for pythiosis has been historically challenging, veterinarians must often resort to other therapies that have shown some degree of success in isolated case reports. In the present study, mefenoxam, prednisone, minocycline, and terbinafine were used in most cases, with hyperbaric oxygen therapy mainly used for cutaneous cases. Although pharmacokinetic studies have not been performed in dogs, mefenoxam has been shown to inhibit *P. insidiosum* in vitro, and it has been demonstrated to be very safe with minimal toxic effects in dogs when given at 8 mg/kg/day [4,5]. In this study, mefenoxam was used at a median dose of 8.1 mg/kg/day (range: 6.7–12.9) with no observed adverse effects. Isolated case reports have shown promise in demonstrating the potential efficacy of this compound in both surgical and non-surgical clinical cases [4]. Response to mefenoxam in cutaneous pythiosis has not been adequately described in previous case reports, and following this retrospective evaluation, cutaneous cases may respond just as well as, if not better than, gastrointestinal cases, particularly when used in conjunction with prednisone, minocycline, and terbinafine.

There are several limitations to this study. First, not all clients consistently updated the clinician on how the dog responded to treatment. This led to one cutaneous and two gastrointestinal dogs being lost to follow-up. These cases could not be included in the final analysis. A definitive time to cure was not available in many cases. Second, many treatments were given concurrently with mefenoxam; therefore, the contribution of each individual treatment to disease progression could not be determined. Prior publications about the treatment of canine pythiosis are limited to case reports and case series. The nature of these previous publications creates some intrinsic limitations about the conclusions that are drawn. For example, causation cannot be inferred from an uncontrolled treatment and subsequent observations; mefenoxam cannot be directly linked to the improved survival of these dogs and thus cannot be generalized to pythiosis treatment [25]. Publication bias is another factor that is considered a common downfall of case reports, as it can lead to presented data that are not truly representative of the disease. Many case reports and case series do not report treatment failures; however, in this study, all dogs, regardless of treatment outcome, were reported if they met the inclusion criteria [25]. Therefore, this potential factor is not considered to be a limiting factor of this study. The lack of a diagnostic test which can confirm complete absence of the causative organism further complicates the study, as does the reliance on referring veterinarian records or owner observation for some cured cases. However, 16 cases were determined to be cured by one of the authors during an examination (n = 11 cutaneous, n = 5 gastrointestinal) which would still support a cure rate of 39% overall, with 44% cutaneous and 31% gastrointestinal, respectively. In the authors’ experience, clinical signs typically recur within three months after remission of lesions, and much of the owner follow-up was obtained long after apparent resolution. Nevertheless, recurrence years after apparent resolution cannot be excluded except with lifetime follow-up. Future controlled studies with more robust controls, regression analysis for multiple factors, and a longer follow-up period should be performed to draw definitive conclusions. 

## 5. Conclusions

In summary, this study suggests that mefenoxam is likely an effective treatment for canine pythiosis when used with other treatments, with minimally-apparent side effects. In addition, this study reinforces that cutaneous pythiosis has a better prognosis (longer survival time) than gastrointestinal pythiosis. Mefenoxam should, therefore, be considered as either a primary therapy or as part of the follow-up adjunctive medical therapy secondary to surgery for the effective treatment of canine pythiosis.

## Figures and Tables

**Figure 1 microorganisms-11-01726-f001:**
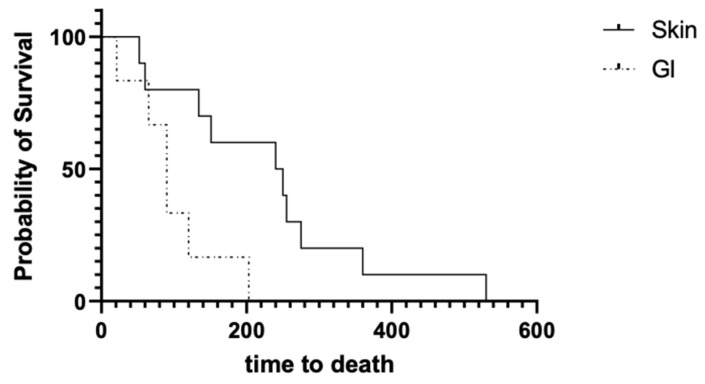
Kaplan–Meier survival curve comparing survival times of the 16 dogs that died from either gastrointestinal (n = 6; dotted line) or cutaneous (n = 10; solid line) pythiosis. Survival time was significantly longer in dogs that died of cutaneous pythiosis than in dogs that died of gastrointestinal pythiosis (Gehan–Breslow–Wilcoxon test, *p* = 0.035).

**Figure 2 microorganisms-11-01726-f002:**
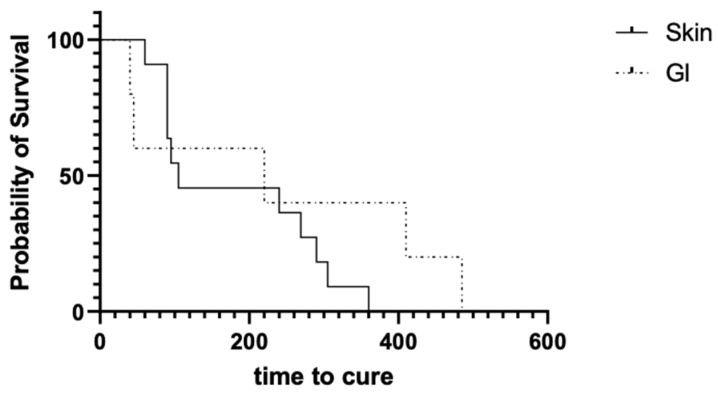
Kaplan–Meier survival curve comparing the amount of time-to-cure in cutaneous (n = 11) versus gastrointestinal (n = 5) pythiosis of the 16 dogs that were confirmed to either have negative serum tests or clinical resolution of signs, excluding dogs for which there was not a definitive time to cure (i.e., resolution of clinical signs could not be traced to a precise date; n = 5). The amount of time for a curative state to be reached did not significantly differ between the two groups (Gehan–Breslow–Wilcoxon test, *p* = 0.95).

**Table 1 microorganisms-11-01726-t001:** Median (range) dosages of mefenoxam and other adjunctive therapies used in combination with mefenoxam. The number of dogs receiving each treatment is reported. HBOT: hyperbaric oxygen therapy. n: number of patients per treatment group.

Cases	Mefenoxam (mg/kg/Day)	Prednisone (mg/kg/Day)	Minocycline (mg/kg/Day)	Terbinafine (mg/kg/Day)	HBOT
Total	8.1 (6.7–12.9), n = 41	0.6 (0.3–2.9), n = 32	10 (5–22), n = 30	10.2 (6.25–61), n = 24	45 min @ 15 psi, n = 23
Cutaneous	8.2 (6.7–12.9), n = 25	0.6 (0.3–2.9), n = 21	10.7 (6.5–22), n = 19	17 (7.5–61), n = 13	45 min @ 15 psi, n = 19
Gastrointestinal	8.0 (7.3–10.2), n = 16	0.6 (0.4–0.9), n = 11	10 (5–14), n = 11	8 (6.25–12.8), n = 11	45 min @ 15 psi, n = 4

**Table 2 microorganisms-11-01726-t002:** Comparison of signalment and clinical values obtained upon presentation of 41 dogs that received mefenoxam after being diagnosed with either gastrointestinal or cutaneous pythiosis between 2013 and 2021. The dogs are subcategorized by disease manifestation of *P. insidiosum* infection as either cutaneous or gastrointestinal. The diagnostic tests performed are listed for each group. BCS = body condition score—nine point scale. Other breeds = Chesapeake Bay Retriever, Great Dane, English Bulldog, and Cur.

Variable	All Dogs (n = 41)	Gastrointestinal (n = 16)	Cutaneous (n = 25)
Age (years) (median; range)	2.5 (0.75–7)	2 (1–6.5)	3 (0.75–7)
Sex			
Male	20	9	11
Female	21	7	14
Breed			
German Shepherd	16	3	13
Labrador Retriever	6	1	5
Siberian Husky	2	2	0
Terrier breeds	2	1	1
Mixed	10	5	5
Other breeds	5	4	1
Weight (kg) (mean ± SD)	-	22.5 (±9.1)	-
BCS (mean ± SD)	**-**	3.5 (±1.3)	**-**
Skin lesions (median; range)	**-**	**-**	1 (1–5)
Diagnostic tests			
Culture	20	2	18
PCR	16	3	13
ELISA	33	12	21
Histopathology	36	12	24
Diagnostic test combinations			
ELISA + culture	1	0	1
ELISA + histopathology	9	5	4
Culture + histopathology	5	2	3
Culture + PCR + histopathology	1	0	1
ELISA + culture + histopathology	4	0	4
ELISA + PCR + histopathology	6	3	3
ELISA + PCR + histopathology + culture	9	0	9

**Table 3 microorganisms-11-01726-t003:** Clinical outcome in infected dogs, median time to death for individuals that died while on mefenoxam, median time to cure for dogs that were definitively free of infection following negative serum tests or examination by a veterinarian. Survivability was calculated by counting all dogs that were alive following cessation of mefenoxam treatment divided by all dogs diagnosed. Death = died or euthanized as a result of infection. Cure = resolution of clinical signs or negative serum test. Lost to follow-up = no information was available in the medical record or by attempted follow-up to determine cure or death. Inconclusive death = euthanasia due to reasons which could not be conclusively linked to pythiosis (e.g., behavioral issues). Time to death = median (range) number of days until death following mefenoxam treatment. Time to cure = median (range) number of days until dogs that survived treatment were considered clinically cured based on ELISA and/or examination by a veterinarian. Note that a time to cure was only available in 11 cutaneous cures and 5 gastrointestinal cures because the remaining cases did not have either an ELISA or an examination at the study institution on a particular date but were otherwise off medications and reportedly free of symptoms. Survivability = cured of disease/[cured of disease + dead from disease + lost to follow-up + inconclusive death].

Cases	Death	Cure	Lost to Follow-Up	Inconclusive Death	Time to Death (Days)	Time to Cure (Days)	Survivability
Total (n = 41)	16	21	3	1	143 (21–530)	163 (40–485)	0.51
Cutaneous (n = 25)	10	13	1	1	245 (52–530)	105 (60–360)	0.52
Gastrointestinal (n = 16)	6	8	2	0	90 (21–203)	220 (40–485)	0.50

## Data Availability

Not applicable.

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
