# Peer review of "The Use of Mefenoxam to Treat Cutaneous and Gastrointestinal Pythiosis in Dogs: A Retrospective Study"

_microorganisms, 2023, doi:10.3390/microorganisms11071726_

Round 1

Reviewer 1 Report

The article is about The use of mefenoxam to treat cutaneous and gastrointestinal pythiosis in dogs: a retrospective study

INTRODUCTION

I suggest that in the Introduction, the authors explain how the dogs get pithiosis, as it´s not common in some countries.

MATERIAL AND METHODS

If the mefexonan is not allowed to use in dogs (the dose is extrapolated – line 76), I think that na ethical citation have to be used in methodology, once the owners had to consent the use of this drug, or not? Please ellucidate it in methodology in any case.

Author Response

Reviewer 1. comments

Point #1. I suggest that in the Introduction, the authors explain how the dogs get pithiosis, as it´s not common in some countries.

Point #2. If the mefexonan is not allowed to use in dogs (the dose is extrapolated – line 76), I think that na ethical citation have to be used in methodology, once the owners had to consent the use of this drug, or not? Please ellucidate it in methodology in any case.

The authors would like to thank the reviewer for this oversight. We have now included the following statement in our methodology Historically, each owner had signed a consent form stating that they were consenting to the use of mefenoxam. Additionally, all owners consented to euthanasia if the disease proved progressive or if the side effects from the mefenoxam affected quality of life.”

Reviewer 2 Report

The study titled “The use of mefenoxam to treat cutaneous and gastrointestinal pythiosis in dogs: a retrospective study” describes the use mefenoxam in various treatment regimens to treat canine pythiosis. This retrospective study relies on medical records to assess the diagnosis and treatment effect in dogs at a single referral veterinary hospital in Florida. The study was done by selecting dogs diagnosed with pythiosis that had received mefenoxam treatment. This makes one question how much this has limited the study as it undoubtedly would have been interesting to see the outcomes of patients with pythiosis that were not treated with mefenoxam. However, the study is interesting and worth publishing after revision. I do however have several concerns that the authors should address, particularly as the authors sometimes over-emphasize the outcomes of the study in the discussion section. Please see more specific comments below.

Line 15: Here it could be specified whether this is a antigen or antibody ELISA.

Line 20-22: If there is room, the authors could add the survival times, e.g. (X vs. Y days).

Line 72-75: The authors claim that several case report and case studies have reported success with mefenoxam treatment, but only provide on case report as reference. Please add more references of change the sentence.

Line 88-90: Why were patients with pythiosis that had not received mefenoxam therapy excluded? Of course, it could be that all patients with this disease received the treatment, but then this would not have to be an inclusion criterion, it could just be reported.

Line 90: Please provide the patient management software used.

Line 93-95: Please provide a reference for the PCR.

Line 106-115: Good description of the statistical analysis!

Line 122-124: Are these the 3+3 dogs mentioned in the previous sentences? Please clarify.

Line 125-126:  Are these the six dogs mentioned in the previous sentence? Pleas clarify.

Table 4 legend: I am not sure that asterisks are needed in the legend as there are no asterisks in the table.

Figure 1: For clarity, please add the number of GI and cutaneous dogs in the legend.

Figure 2: For clarity, please add the number of GI and cutaneous dogs in the legend.

Line 262-265: I am not sure the authors can claim this: “…the results of this study show that dogs with gastrointestinal or cutaneous pythiosis have an increased change of survival… …when mefenoxam is given orally in conjunction with other therapeutic interventions.” The study does not have a control group that did not received mefenoxam, thus the role of the drug cannot be evaluated. The authors state this very thing on lines 355-358. Please rephrase.

The above is also the case later in the discussion; I suggest the authors be very cautious when claiming that their study shows that treatment with mefenoxam increases life survival time of dogs with pythiosis compared with treatment without it. Clear statements of superiority of one treatment regiment compared to another would require a robust placebo-controlled superiority trial. The authors do point out, rightfully so, that many factors may affect the outcome of the disease, hence the need for a prospective trial.

Line 366-368: Here it should be made clear that the prognosis is better in cutaneous cases only with the therapy described, not in general.

Author Response

Point #1. Line 15: Here it could be specified whether this is a antigen or antibody ELISA.

  • The authors would like to thank the reviewer. We have made the appropriate change the description of the ELISA to now read “Dogs were included in this study upon previous definitive diagnosis through the organism identification via culture, PCR, antibody ELISA, post-mortem identification, or a combination of these tests with or without supportive histopathological analysis.”

Point #2. Line 20-22: If there is room, the authors could add the survival times, e.g. (X vs. Y days).

  • The author would like to thank the reviewer for their suggestion and have added in specific data about survival times. The sentence now reads “There was a statistically significant difference in the time to death between cutaneous (245 days (52 - 530)and gastrointestinal (90 days (21 - 203) groups; dogs infected with cutaneous pythiosis survived significantly longer after being diagnosed with the disease (p=0.035).”

Point #3.  Line 72-75: The authors claim that several case report and case studies have reported success with mefenoxam treatment, but only provide on case report as reference. Please add more references of change the sentence.

  • The authors would like to thank the reviewer for drawing our attention to this oversight. We have added further references to strengthen this statement.

·         Pornphan Sukanan, Bongkot Suparp, Supattra Yongsiri, Piyarat Chansiripornchai, Sawang Kesdangsakonwut Successful management of colonic pythiosis in two dogs in Thailand using antifungal therapy Vet Med Sci 2022 Nov;8(6);2283 - 2291

  • Harry Cridge, MVB, MS, MRCVS, Samantha M. Hughes, BA, DVM, Vernon C. Langston, DVM, PhD, DACVCP, Andrew J. Mackin, BVMS, DVSc, DACVIM Mefenoxam, Itraconazole, and Terbinafine Combination Therapy for Management of Pythiosis in Dogs (Six Cases) J Am Anim Hosp Assoc(2020) 56 (6): 307.

Point #4. Line 88-90: Why were patients with pythiosis that had not received mefenoxam therapy excluded? Of course, it could be that all patients with this disease received the treatment, but then this would not have to be an inclusion criterion, it could just be reported.

  • Patients were excluded from the study of they did not receive mefenoxam because the study is specifically about the use of mefenoxam to treat cutaneous and gastrointestinal pythiosis in dogs.

Point #5. Line 90: Please provide the patient management software used.

  • The authors have added the patient management software used “Electronic medical records (Cornerstone® Veterinary Software, IDEXX®) of all dogs examined, diagnosed with pythiosis, and treated with mefenoxam were searched between the years 2013 and 2020 at the tertiary referral center.”

Point #6.  Line 93-95: Please provide a reference for the PCR.

·         A reference has been added for the PCR

o   Nadine R. Znajda, Amy M. Grooters, Rosanna Marsella PCR-based detection of Pythium and Lagenidium DNA in frozen and ethanol-fixed animal tissues. Veterinary Dermatology 2002, 13, 187– 194

Point #7. Line 122-124: Are these the 3+3 dogs mentioned in the previous sentences? Please clarify.

  • The authors apologize for the confusion. The entire section has been edited to increase clarity.

Point #8: Line 125-126:  Are these the six dogs mentioned in the previous sentence? Pleas clarify

  • The authors apologize for the confusion. The entire section has been edited to increase clarity.

Point #9. Table 4 legend: I am not sure that asterisks are needed in the legend as there are no asterisks in the table.

  • The asterisks have been removed as per the reviewer suggestions.

Point #10. Figure 1: For clarity, please add the number of GI and cutaneous dogs in the legend.

  •  

Point #11. Figure 2: For clarity, please add the number of GI and cutaneous dogs in the legend.

  •  

Point #12. Line 262-265: I am not sure the authors can claim this: “…the results of this study show that dogs with gastrointestinal or cutaneous pythiosis have an increased change of survival… …when mefenoxam is given orally in conjunction with other therapeutic interventions.” The study does not have a control group that did not received mefenoxam, thus the role of the drug cannot be evaluated. The authors state this very thing on lines 355-358. Please rephrase.

The above is also the case later in the discussion; I suggest the authors be very cautious when claiming that their study shows that treatment with mefenoxam increases life survival time of dogs with pythiosis compared with treatment without it. Clear statements of superiority of one treatment regiment compared to another would require a robust placebo-controlled superiority trial. The authors do point out, rightfully so, that many factors may affect the outcome of the disease, hence the need for a prospective trial.

  • The authors would like to thank the reviewer for their insightful comments. The authors have now amended their previous statement to be in line with our actual conclusions from the study. “Based on our study, mefenoxam given orally in conjunction with other therapeutic interventions resulted in survival times of 245 days (52 - 530)and 90 days (21 - 203) for cutaneous and gastrointestinal disease respectively.  Previous publications regarding the use of mefenoxam in cases of canine pythiosis are limited, with treatments being highly variable, therefore no direct comparisons can be made about survival in the current study and previous case series or single case reports.”   

Point #13. Line 366-368: Here it should be made clear that the prognosis is better in cutaneous cases only with the therapy described, not in general.

  • The authors would like to thank the reviewer for their comments. We have now amended the sentence to read as follows; Cutaneous pythiosis response to mefenoxam has not been adequately described in previous case reports, and following this retrospective evaluation, cutaneous cases may respond just as well, if not better than gastrointestinal cases with combination therapy consisting of mefenoxam, prednisone, minocycline, and terbinafine.”
